# Brief communication: Hydrological and hydraulic investigation of the extreme September 2024 flood on the Lamone River in Emilia-Romagna, Italy

Alessia Ferrari[1], Giulia Passadore[2], Renato Vacondio[1], Luca Carniello[2], Mattia Pivato[2], Elena Crestani[2], Francesco Carraro[2], Francesca Aureli[1], Sara Carta[1], Francesca Stumpo[3], Paolo Mignosa[1]

[1]Department of Engineering and Architecture, University of Parma, Parma, 43124, Italy
[2]Department of Civil Environmental and Architectural Engineering, University of Padova, Padova, 35131, Italy
[3]Civil Protection Agency of Emilia Romagna, Bologna, 40122, Italy

*Correspondence to*: Alessia Ferrari (alessia.ferrari@unipr.it)

**Abstract.** In September 2024, several European countries experienced extreme and prolonged record-breaking rainfall that induced severe flooding and caused widespread damage, casualties and disruptions. In this context, the Emilia-Romagna Region in Northern Italy suffered heavy precipitation primarily affecting the Lamone River basin, where a levee breach occurred near the village of Traversara, causing the flooding of urban settlements, vineyards, orchards and crops. Since the same area was severely impacted by devastating floods no later than May 2023, it is relevant to understand whether this area indeed faced extreme precipitation events in two consecutive years and to explore how the hydrological-hydraulic modelling can support the preparedness against these recurring events.

## 1 Introduction

Between September 17 and 20, 2024, the eastern part of the Emilia-Romagna Region in Northern Italy, a highly economically developed area that already suffered extensive and devastating floods in May 2023 due to the overflowing of 23 rivers (Arrighi & Domeneghetti, 2023), was hit by another episode of extreme precipitations. This intense rainfall event was driven by the Mediterranean storm Boris, which on September 11th moved eastwards from the Gulf of Genoa, reaching central Europe and causing widespread flooding in several countries before weakening (ARPAE, 2024). By September 17th, the storm intensified its strength, likely exacerbated by the summer warming trend of the temperatures along the Adriatic Sea (ARPAE, 2024), resulting in severe hydrological and meteorological conditions that impacted the eastern part of the Emilia-Romagna Region. Due to this severe rainfall event, on September 19th 2024, a breach formed along the left levee of the Lamone River near the Traversara village, causing the flooding of a large area between the cities of Imola and Ravenna. The downstream stretch of the Lamone River, like most of the rivers of this region, is characterised by the presence of artificial earthen levees of relevant height above the nearby lowland, and the riverbed itself often lies higher than the surrounding ground level. After a few hours of overflow, a 40-meter section of the left levee collapsed at about 11:30 a.m., triggering widespread flooding. Elsewhere

along the river, no levee failures occurred. Both the rainfall event and the consequent flooding were widely perceived by the media as yet another exceptional event impacting areas still recovering from the 2023 flood.

To better assess the severity of this event, this brief communication aims at (i) evaluating the main characteristics of the hydrological event against historical records and (ii) investigating the flood impacts by reconstructing the inundation combining the use of both a hydrological and a hydrodynamic numerical model.

## 2 Characterisation of the rainfall event

The hourly precipitations recorded during the considered event were extracted from the high-resolution ERG5-Eraclito dataset, which provides historical precipitation and temperature observations on a $5 \times 5$ km cell grid covering the entire Emilia-Romagna Region (Antolini et al., 2016). The accumulated rainfall was evaluated from midnight on September 17 to 11:00 p.m. on September 20 and spatially interpolated using the Shepard algorithm (consistently with the method adopted by the ERG5-Eraclito dataset). As shown in the resulting rainfall map in Fig. 1, the highest precipitations fell inside the Lamone River basin (around 520 km$^2$), which includes the Lamone River and its right tributary, the Marzeno River, that flows into the main course just upstream of the town of Faenza. Over these two days, the river basin was affected by a rainfall amount of about 270 mm, with 320 mm concentrated within an area of around 270 km$^2$ defined by the 280 mm isohyet (Fig. 1): the average precipitation depth in the central portion of the basin was approximately 1.8 times higher than the mean precipitation over the eastern part of the region.

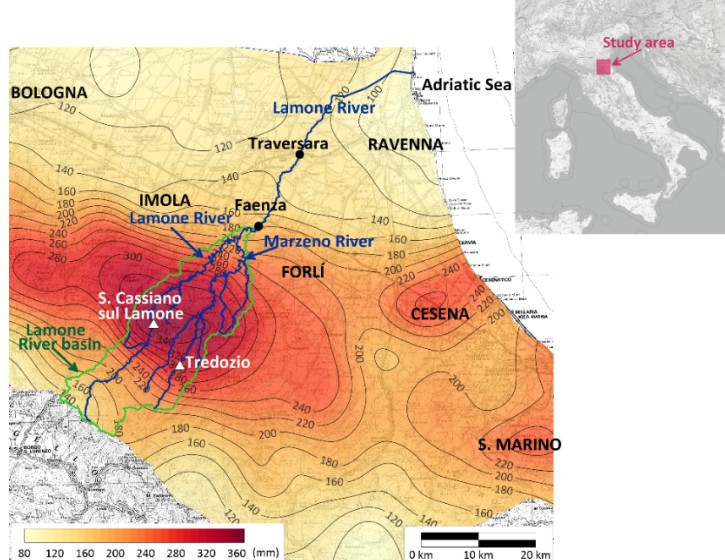

**Figure 1. Rainfall map of the September 17-20, 2024 event on the eastern part of the Emilia-Romagna Region in Northern Italy. The Lamone River basin, closed in the town of Faenza, is depicted. The triangles indicate the rain gauge stations considered for the statistical analysis. The image in the background is made available by Emilia-Romagna Region at https://geoportale.regione.emilia-romagna.it/ (last access: 2 December 2024).**

To statistically assess the severity of this event within the most affected river basin, the cumulative rainfall at specific stations was evaluated for the durations of 1, 3, 6, 12, and 24 hours and the resulting values were compared with the annual maxima recorded in the period 1928-2023 (Table 1). This analysis focused on the rain gauges of San Cassiano sul Lamone in the Lamone catchment (95 years of records) and Tredozio in the Marzeno catchment (43 years of records) (Fig. 1) due to their extensive observation period. At both rain gauges, the rainfall accumulation over 24 hours broke the historical records by doubling the highest values reached in the period 1928-2022 and being nearly 1.5 times higher than the May 2023 ones. Even for the durations of 3, 6, and 12 hours, the rainfall accumulations of the considered event at the San Cassiano sul Lamone rain gauge overcame the historical ones from 1.6 to 1.9 times. In contrast, at the Tredozio rain gauge in the Marzeno catchment, the 2024 event exceeded the historical records by about 1-1.4 times (Table 1). It is interesting to note that at the considered rain gauges, the 2024 event overcame the 2023 one, even if its catastrophic impacts affected smaller areas than in May 2023.

**Table 1. Highest rainfall accumulations evaluated over 1 ($h_1$), 3 ($h_3$), 6 ($h_6$), 12 ($h_{12}$), and 24 ($h_{24}$) hours at the considered rain gauges on the Lamone and Marzeno Rivers during the September 2024 event, the extreme events of May 2023 and the historical dataset 1928-2022. For each station and duration, the ratios $r$ between the rainfall accumulation resulting from September 17-20, 2024 and the other ones are reported.**

| Station | Period | $h_1$ (mm) | $r_1$ (-) | $h_3$ (mm) | $r_3$ (-) | $h_6$ (mm) | $r_6$ (-) | $h_{12}$ (mm) | $r_{12}$ (-) | $h_{24}$ (mm) | $r_{24}$ (-) |
|---|---|---|---|---|---|---|---|---|---|---|---|
| **San Cassiano sul Lamone** (Lamone River) | September 2024 | 52 | - | 120 | - | 166 | - | 224 | - | 288 | - |
| | May 2023 | 17 | 3.1 | 41 | 2.9 | 71 | 2.3 | 125 | 1.8 | 208 | 1.4 |
| | 1928-2022 | 56 | 0.9 | 77 | 1.6 | 96 | 1.7 | 118 | 1.9 | 151 | 1.9 |
| **Tredozio** (Marzeno River) | September 2024 | 33 | - | 81 | - | 115 | - | 147 | - | 215 | - |
| | May 2023 | 24 | 1.4 | 43 | 1.9 | 73 | 1.6 | 116 | 1.3 | 169 | 1.3 |
| | 1928-2022 | 58 | 0.6 | 84 | 1.0 | 100 | 1.2 | 110 | 1.3 | 114 | 1.9 |

The possible presence of trends in the historical observations was detected by adopting the Mann-Kendall test (Mann, 1945; Kendall, 1970) with a 5% significance level. This non-parametric test, which is widely used for hydro-meteorological data analysis, statistically assesses potential upward or downward trends over time by comparing the number of concordant pairs of data points. For both the selected rain gauges, the resulting tendencies are not statistically significant. Moreover, the Pettitt test (Pettitt 1979), which adopts a non-parametric approach to detect change points in the time series, proved that the historical datasets are homogeneous for the considered 5% significance level.

The probability distributions capable of describing the samples were selected according to the $L$-moments method (Hosking & Wallis, 1997). The $L$-skewness ($L_3$) and $L$-kurtosis ($L_4$) moments were evaluated for each rainfall dataset, and the five resulting pairs were plotted on the $L$ moment ratio diagram, together with their mean value (Fig. 2a, b).

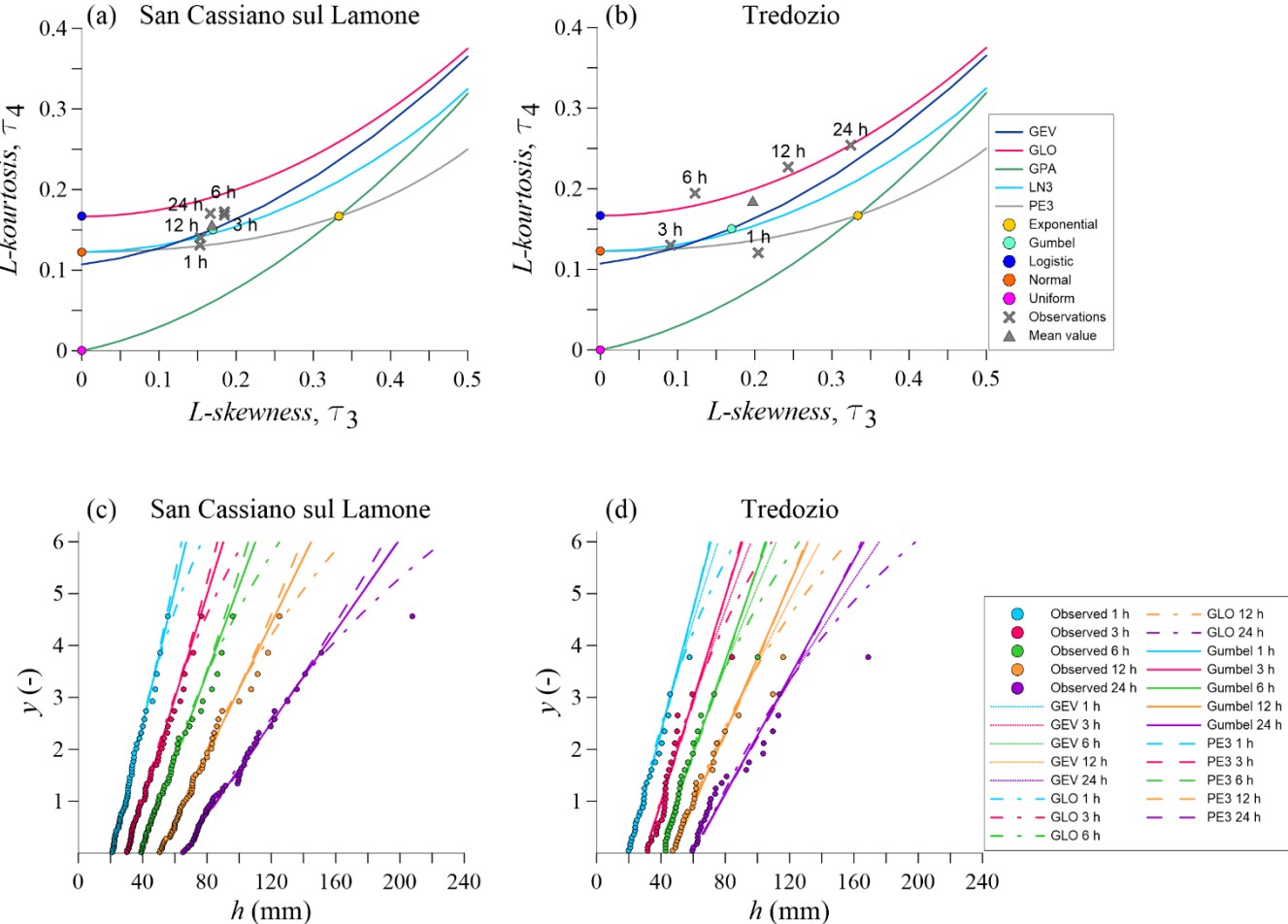

**Figure 2. (a, b)** *L* moment ratio diagrams and **(c, d)** probability charts for the rain gauges of **(a, c)** San Cassiano sul Lamone and **(b, d)** Tredozio. In **(c)**, the GEV distribution lines overlap the Gumbel ones.

As suggested by the diagrams in Fig. 2, the following theoretical distributions were considered: Generalized Extreme Value (GEV), Generalized Logistic (GLO), 3-parameters Lognormal (LN3), Pearson type 3 (PE3) and Gumbel. Beyond the visual inspection, the model selection was assessed by evaluating the Akaike (AIC) and the Bayesian (BIC) information criterion, the Consistent AIC (CAIC) and the Hannan-Quinn (HQC) criterion. The resulting indicators suggested that the Gumbel distribution was the best suited for describing the datasets: e.g. for the San Cassiano sul Lamone rain gauge, the differences between the AIC resulting from the Gumbel distribution and those resulting from the PE3, the GEV, the LN3, and the GLO were around 3, 2, 2 and 4, respectively. Thus, the Gumbel distribution was selected to estimate the return period of the accumulated precipitations recorded in September 2024. At the San Cassiano sul Lamone station, the 1-hour cumulative rainfall depth corresponded to a return period of approximately 55 years, while the 3, 6, 12 and 24-hour rainfall exceeded the 500-year return period. Similarly, at the Tredozio rain gauge, the 1-hour and 3-hour precipitations were associated with return

periods of about 5 years and 150 years, respectively, whereas the 6, 12, and 24-hour accumulations exceeded the 500-year return period.

## 3 Modelling of the hydrological response

The rainfall-runoff processes occurring over the Lamone River watershed were analysed adopting the Rhyme (River
HYdrological ModEl) hydrological model, a spatially explicit model that discretizes hydrological processes into each sub-catchment of the basin and then routes flow exiting from each sub-catchment along the digitalised channel network (Rinaldo, A. & Rodriguez-Iturbe, 1996). The rainfall-runoff model configuration is largely based on the version described by Schaefli et al. (2014), which accounts for rainfall/snowfall separation, interception and re-evaporation of intercepted water, snowpack evolution and equivalent precipitation-runoff transformation, with some minor modifications, i.e. the exclusion of interception
and glacier runoff components, and the inclusion of urban runoff generated from developed areas. The runoff generation is controlled by both infiltration and saturation excess processes. The model was driven by hourly rainfall recorded at 38 meteorological stations that were firstly interpolated and then mediated in each sub-catchment, daily cumulative potential evapotranspiration and daily average temperatures, which were still recovered from the high-resolution ERG5-Eraclito dataset (Antolini et al., 2016): daily average temperatures were used to estimate snowmelt while daily potential evapotranspiration
was incorporated into the root zone water balance. In addition to the nonlinear root zone reservoir, water storage in each sub-catchment was represented by four additional linear reservoirs (urban, superficial, sub-superficial and deep): the recession times of these reservoirs were assumed to be uniform across all sub-catchments independently of their areas. Meteorological forcing was assumed to be uniformly distributed within each sub-catchment, whose definition was delineated using a digital elevation model and the hydrologic analysis tools available in ArcGisPro software to determine flow direction, calculate flow
accumulation, delineate watersheds, and create stream networks.

Past observations of streamflow collected at Reda gauging station (located at the outlet of the river basin) from January 2008 till October 2024 were used to estimate model parameters through the Differential Evolution Adaptive Metropolis (DREAMZS) implementation of the Markov Chain Monte Carlo algorithm (MCMC) (Vrugt et al., 2009). Given the prior probability density function of the hydrological model parameters requiring calibration and the available observations,
DREAMZS samples the desired number of parameter realisations from the posterior distribution using multiple MCMC chains that run in parallel and jointly contribute to the computation of the proposed parameter samples. The likelihood function is computed assuming independent, identically distributed normal errors between observed and simulated discharge values. As a result, the calibration procedure enabled the estimation of the hydrological model parameters (such as maximum infiltration capacity, snowmelt temperature and reservoir residence time), which were assumed as uniformly distributed across the 46 sub-
catchments defining the Lamone River basin closed at the Reda gauging station, and the reconstruction of the flood waves occurred over the last 16 years (Fig. 3). Since the long time series of discharges adopted in the calibration step was actually recovered through rating curves from the stage hydrographs recorded at the Reda gauging station, the extreme flood events of

May 2023 and September 2024 were not included in the calibration dataset due to the uncertainties in the available stage-discharge relationships. The calibration of the hydrological model was quantitatively assessed by using the Nash-Sutcliffe
Efficiency indicator defined as:

$$NSE = 1 - \frac{\sum_{i=1}^{N}(Q_{m,i} - Q_{o,i})^2}{\sum_{i=1}^{N}(Q_{o,i} - \bar{Q}_o)^2} \qquad (1)$$

where $Q_{m,i}$ is the $i$th value of the modelled discharges, $Q_{o,i}$ is the $i$th value of the observed discharges, and $\bar{Q}_o$ is the mean value of the observed discharges.

As shown in Fig. 3, the hydrological model well reproduced the 2008-2024 discharge series and even the September 2024 flood wave, which represented the second most severe event ever occurred in this station: the NSE indicator resulted equal to
130 0.83, further demonstrating the model's performance.

Due to the availability of a calibrated hydrological model, the flow hydrographs resulting on the Lamone and Marzeno Rivers upstream of the town of Faenza were extracted and provided as boundary conditions to the hydrodynamic model (Fig. 3). Since neither minor channels nor tributaries flow into the Lamone River downstream of the town of Faenza, no additional inflows were considered in the hydraulic modelling.

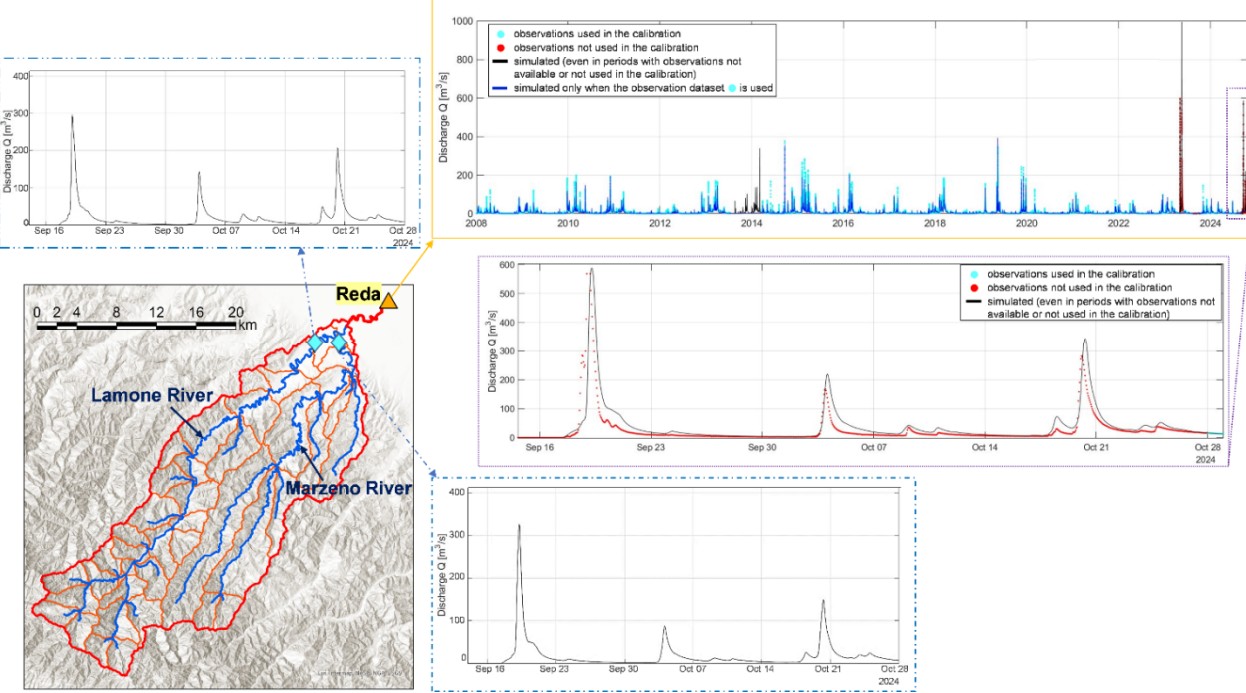

**Figure 3. Calibration of the hydrological model. The comparison between the 2008-2024 series of discharges observed and simulated at the Reda gauging station, with a focus on the considered event, is depicted. During the calibration, the extreme events of May 2023 and September 2024 were excluded from the calibration dataset (red circles), while the reconstructed discharges were simulated both in the presence (blue line) and in the absence (black line) of observations. The flow hydrographs resulting on the Lamone and**
140 **Marzeno Rivers to be imposed as upstream boundary conditions in the hydrodynamic modelling are also represented (light-blue diamonds). The river watershed and its 46 sub-basins are finally shown.**

## 4 Simulation of the flooding event

The flooding dynamic was investigated by simulating the levee-breach-induced inundation with the PARFLOOD numerical model, which is a two-dimensional finite volume scheme parallelised on Graphic Processing Units (GPUs) (Vacondio et al.,

2017). The study domain, which included a 60 km stretch of the Lamone River from Faenza to the Adriatic Sea, 1 km of the Marzeno River and 700 km$^2$ of floodable areas, was described using a 1 m × 1 m LiDAR-based DTM, surveyed in 2023. The computational grid was defined by down-sampling the resolution up to 4 m meanwhile preserving the crest elevation of all the embankments spread in the domain (Ferrari et al., 2020). Concerning the model setup, the river roughness was retrieved from literature values (Chow, 1959), whereas in the lowlands, the values defined in Ferrari et al. (2020) were assumed. As upstream

boundary conditions, the discharge hydrographs resulting from the hydrological model (Sect. 3) were imposed (upstream of the town of Faenza) on the Lamone and Marzeno Rivers, respectively, while a constant sea water level equal to 0 m a.s.l. was imposed as downstream boundary condition at the river mouth into the Adriatic Sea (Fig. 4).

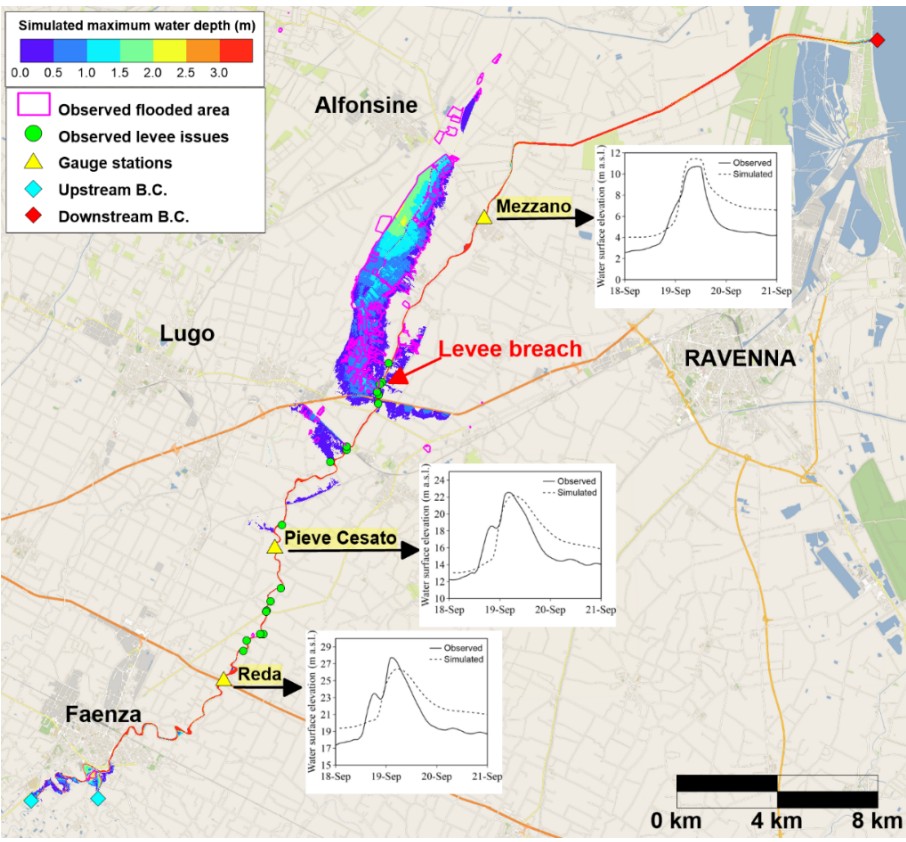

**Figure 4. Resulting maximum water depths and boundary of the observed flooded area. The comparison between the water levels**
**recorded during the event at three gauge stations and those simulated is also depicted. The location of the upstream and downstream boundary conditions is finally indicated. Background map: © OpenStreetMap contributors (2024), distributed under the Open Data Commons Open Database License (ODbL).**

The breach opening on the left bank of the Lamone River close to the Traversara village was modelled adopting a geometric approach: according to direct observations, at 11.30 a.m. of September 19$^{th}$, the breach started deepening, from the levee crest up to the ground level, and widening, reaching the maximum width of 40 m a few hours later. The numerical results show that the levee-breach-induced flood hit the village of Traversara first, and in less than 10 hours, urban settlements spread in the lowlands, crops, vineyards, and orchards. The presence of minor channel embankments to the west confined the flood propagation, preventing the flooding of the Alfonsine village (around 12000 inhabitants). Moreover, some localized levee overtopping and related issues occurred upstream of the breach location, as confirmed by local authorities (Fig. 4). The simulated flooded area closely matches the actual one and the comparison between the water levels predicted by the model and those observed at three gauge stations fairly agrees in terms of timing and pick values (at Reda, Pieve Cesato and Mezzano stations, the misfits relative to the peak values are lower than the 5%, 2% and 9%, respectively). Some discrepancies in the falling limbs can be mainly ascribed to the fact that the hydrological model does not account for levee overtopping upstream of the Reda station and that the hydrodynamic model slightly overestimates low water levels. This overestimation arises since the riverbed geometry in the DTM was recovered from a LiDAR survey carried out under non-drought conditions. Nonetheless, it is important to highlight that the hydrological-hydraulic modelling effectively reconstructed this recent flooding event, and that the hydrodynamic simulation ensured a ratio of physical to computational times of about 20.

## 5 Closing remarks

Over the past two years, the Lamone River in Northern Italy experienced two extreme hydrological events. The preliminary investigation carried out in this brief communication showed that within the eastern part of the Emilia-Romagna Region, the Lamone River basin was the area most severely affected by the rainfall event of September 2024. The cumulative rainfall evaluated at two stations within this area broke the long historical observation dataset by doubling the highest values recorded in the period 1928-2022. The statistical analysis of the accumulated precipitations over 6, 12 and 24 hours proved that the return period of this event exceeded 500 years. Moreover, it is worth noting that even the return period of May 2023 events, which was evaluated by carrying out the statistical analysis described in Sect. 2 considering the 1928-2022 observation dataset, was estimated to exceed the 500 years return period for the precipitations accumulated over 24 hours (for the remaining durations the return periods were in the range 2-250 years). While it is quite unexpected for independent extreme and rare events to occur in consecutive years, this is not the case for the 2023 and 2024 events at the considered stations: occurring 16 months apart, they are unrelated in terms of both weather conditions and of initial soil moisture.

The statistical estimates were based on the assumption of stationarity, i.e. past data can be used to predict future hydrological behaviour, a common approach in hydrological analysis. However, this assumption may no longer hold due to climate change, as future events might not follow historical patterns. For instance, climate projections for the Emilia-Romagna Region suggest an increase in future precipitation, also indicating a possible influence of climate change in growing the intensity of recent events (Ginesta et al., 2024).

The severity of the 2024 hydrological event was also confirmed by the maximum water levels recorded at three gauge stations along the Lamone River. The Reda station, located downstream of the confluence between the Lamone and the Marzeno Rivers, registered a maximum water level of 11.4 m, 3 m above the 1999-2022 record (in May 2023, the maximum observed water level was 11.8 m). Even at the downstream stations of Pieve Cesato and Mezzano, the maximum water levels of the 2024 event were close to those recorded in 2023, thus breaking the historical records.

This severe rainfall event caused significant damage to a highly exposed area characterised by extensive industrial and agricultural activities, as well as valuable environmental and historical assets. Not surprisingly, according to the latest report of the Italian National Institute for Environmental Protection and Research (ISPRA, 2021), the 45% of the population living in the Emilia-Romagna Region is exposed to flood risk for both low and medium flood frequency (against an average national value equal to the 15%). Within this context, the reconstruction of the September 2024 event has further highlighted the

capability of numerical models to capture the complex interactions between floods and urban environments. Particularly, the adoption of hydrological and hydraulic models, such as Rhyme and PARFLOOD, can facilitate the assessment of extreme events, thus providing insights for increasing the preparedness for at-risk populations, both through off-line analyses and real-time predictions.

**Acknowledgements**

AF acknowledges the CINECA award under the ISCRA initiative, for the availability of high-performance computing resources and support, and she also acknowledges that this research was granted by University of Parma through the action Bando di Ateneo 2024 per la ricerca. RV and PM acknowledge financial support from the PNRR MUR project ECS_00000033_ECOSISTER. This research benefits from the MarghERita cluster of the Emilia-Romagna Region and the HPC (High Performance Computing) facility of the University of Parma, Italy. The editor, the anonymous reviewer, and Dr.

Gordon Woo are also kindly acknowledged for their valuable suggestions on the early version of the manuscript.

**Author contributions**

Conceptualization: AF. Formal analysis and methodology: AF, GP. Software: AF, GP, RV. Supervision: RV, LC, PM. Field data provision: FS. Writing-original draft preparation: AF with contributions of GP. All authors reviewed the final paper.

**Competing interests**

The authors declare that they have no conflict of interest.

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
