# Peer review of "Brief communication: Hydrological and hydraulic investigation of the extreme September 2024 flood on the Lamone River in Emilia-Romagna, Italy"

_EGUsphere, 2025_

## Author Response (AR1)

**RESPONSE TO THE COMMENTS OF REVIEWER #1 DR. GORDON WOO**

**This is an interesting and important brief communication on flooding in the Lamone River Basin.**

We thank Dr. Woo for his appreciation and positive feedback on our work.

**The closing remarks section includes the following statement:**

*As a matter of fact, over the last two years, this region has been affected by two rainfall events that, at least for the considered stations, exceeded the 500-years return period, despite they occurred in different seasons (Spring and Autumn) and presented different characteristics (two consecutive events in May 2023 and a unique event in September 2024).*

**The authors should provide some interpretive explanation for these exceedances. To what extent have the high return periods been exaggerated because they have not taken adequate account of climate change? Maybe in the current climate, they are not so rare.**

We thank the Referee for raising this interesting point, and we agree that this statement requires more explanation. The statistical analysis described in Sect. 2 enabled us to associate a return period with a given rainfall event following the assumption of stationarity, which is commonly adopted to define future hydrologic information based solely on historical data. Thus, the return period for the May 2023 events was estimated using the 1928-2022 observation dataset, whereas the return period for the 2024 event was evaluated using the 1928-2023 sample. To better assess the severity of the 2024 rainfall, the statistical analysis was repeated by excluding the 2023 event from the database (thus still using the 1928-2022 time series): in both cases, the resulting return periods for accumulated precipitation over 24 hours still exceeded 500 years.

The concept of stationarity represents one of the main challenges when dealing with future events since it is threatened, for example, by climate change's role. Rare past events may become less rare in the present and future climate. However, while quantifying the effects of climate change on small river basins is still challenging, recent evaluations at the European scale highlighted the role of climate projections in increasing the severity and frequency of present and future rainfall events (e.g. Alfieri et al., 2015). Moreover, focusing on the Emilia-Romagna region, a recent study analysing the May 2023 events detected the role of climate change in increasing the severity of recent cyclones (Ginesta et al., 2024).

In the revised paper, we have modified the description on page 8, lines 180-183, to better emphasise the evaluation of the return period for the 2023 events as follows:

*"Moreover, it is worth noting that even the return period of May 2023 events, which was evaluated by carrying out the statistical analysis described in Sect. 2 considering the 1928-2022 observation dataset, was estimated to exceed the 500 years return period for the precipitations accumulated over the 24 hours (for the remaining durations the return periods were in the range 2-250 years)."*

Moreover, we have described the concept of stationarity and discussed the role of climate change on page 8, lines 185-189 as follows:

*"The statistical estimates were based on the assumption of stationarity, i.e. past data can be used to predict future hydrological behaviour, a common approach in hydrological analysis. However, this assumption may no longer hold due to climate change, as future events might not follow historical patterns. For instance, climate projections for the Emilia-Romagna Region suggest an increase in future precipitation, also indicating a possible influence of climate change in growing the intensity of recent events (Ginesta et al., 2024)."*

Alfieri, L., Feyen, L., Dottori, F., & Bianchi, A. (2015). Ensemble flood risk assessment in Europe under high end climate scenarios. Global Environmental Change, 35, 199-212.

Ginesta, M., Lu, C., Coppola, E., Yiou, P., and Faranda, D.: Climate change fingerprint on the 2023 Emilia Romagna floods, EMS Annual Meeting 2024, Barcelona, Spain, 1–6 Sep 2024, EMS2024-448, https://doi.org/10.5194/ems2024-448, 2024.

**For two independent rare events to occur in consecutive years is, a priori, very unlikely. To what extent are the extreme rainfall events in 2023 and 2034 actually connected, i.e. through antecedent conditions etc.?**

We thank the Referee for this comment, as it is an excellent point to discuss. We agree that the occurrence of independent and extreme events in consecutive years is quite unexpected. However, the hydrological characterisation confirmed this behaviour for the considered stations. For both stations, the estimated return periods associated with 24-hour accumulated precipitation exceed 500 years, thus confirming the exceptionality of the 2023 and 2024 events. Moreover, the 2024 event occurred 16 months after the 2023 one, a long time interval that ensures the independence of the events and excludes any correlation either in terms of weather conditions or of initial soil conditions.

In the revised paper, on page 8, lines 182-184, we have clarified this aspect as follows:

*"While it is quite unexpected for independent extreme and rare events to occur in consecutive years, this is not the case for the 2023 and 2024 events at the considered stations: occurring 16 months apart, they are unrelated in terms of both weather conditions and of initial soil moisture."*

**RESPONSE TO THE COMMENTS OF REVIEWER #2**

**This first reconstruction presents valuable insights and thoroughly addresses the multiple components typically required in a first analysis of a flood event—from the use of statistical rainfall data to hydrological modelling and the subsequent forcing of a hydrodynamic model. The manuscript is generally well structured, clear, and pleasant to read. The overall simulation strategy also appears coherent.**

We thank the Referee for his/her appreciation of our work.

**Based on the analysis of this brief communication, several questions and comments arise. Points 1 and 2 are deemed essential for ensuring methodological transparency and simulation reproducibility. Points 3 to 5 raise global questions for potential further analysis or clarification, and points 6 and 7 pertain to minor remarks and suggestions for additional clarification or enhancement.**

We thank the Referee for his/her detailed comments and suggestions.

1. **The rainfall-runoff model employed ('Rhyme') is not sufficiently described. The citation of Rinaldo et al. (1996) does not provide enough information to allow readers to understand or reproduce the model. Although temperature and evapotranspiration data are mentioned as model inputs, the manner in which they are used, and whether they are coupled or treated independently, remains unclear. It would be beneficial to include a conceptual schematic of the model to illustrate the interaction between storage reservoirs and their spatial configuration (e.g. whether all reservoirs are used per sub-catchment or if some of them are shared across the domain). If further references or documentation exist, they should be cited explicitly.**

    We thank the Referee for this comment. In the revised paper, we have provided more details related to the rainfall-runoff scheme of the Rhyme model and added a new reference to the work of Schaefli et al. (2014) as follows (page 5 lines 97-101):

    *"The rainfall-runoff model configuration is largely based on the version described by Schaefli et al. (2014), which accounts for rainfall/snowfall separation, interception and re-evaporation of intercepted water, snowpack evolution and equivalent precipitation-runoff transformation, with some minor modifications, i.e. the exclusion of interception and glacier runoff components, and the inclusion of urban runoff generated from developed areas. The runoff generation is controlled by both infiltration and saturation excess processes."*

    Moreover, at page 5 lines 104-105 we have clarified the manner in which daily average temperatures and potential evapotranspiration are used:

    *"daily average temperatures were used to estimate snowmelt while daily potential evapotranspiration was incorporated into the root zone water balance."*

    Schaefli, B., Nicótina, L., Imfeld, C., Da Ronco, P., Bertuzzo, E., and Rinaldo, A.: SEHR-ECHO v1.0: a Spatially Explicit Hydrologic Response model for ecohydrologic applications, Geosci. Model Dev., 7, 2733–2746

2. **It is unclear which objective function(s) were used in the MCMC calibration algorithm. Please specify the criteria used to assess the model fit during calibration.**

    We thank the Reviewer for this suggestion. In the revised paper (page 5, lines 116-117) we have clarified this point as follows:

*"The likelihood function is computed assuming independent, identically distributed normal errors between observed and simulated discharge values."*

3.  **The model includes linear reservoirs, suggesting that the recession time parameter may depend on sub-catchment area. As their surface can strongly vary and all parameters are un, this might have a significant impact. Was this parameter treated in a dimensionless way, for instance, scaled by surface area or other global catchment characteristics?**

    We thank the Reviewer for this comment. In the revised paper, we have better specified the linear reservoirs adopted by the model as follows (page 5, lines 105-107):

    *"In addition to the nonlinear root zone reservoir, water storage in each sub-catchment was represented by four additional linear reservoirs (urban, superficial, sub-superficial and deep): the recession times of these reservoirs were assumed to be uniform across all sub-catchments independently of their areas."*

4.  **The good performance of the hydrological model in 2024 and the overestimation of discharges in Reda during 2023, is somewhat surprising. Given that the model was calibrated on historical series dominated by moderate and low flows, one might have expected peak underestimation rather than overestimation. Could the authors provide possible causes for these results?**

    We thank the Referee for having raised this interesting point. The hydrological model aimed to infer inflow hydrographs for the Lamone and Marzeno Rivers upstream of their confluence. However, during the 2023 event, several overflowing occurred between this confluence and the Reda station. Thus, the "observed" discharge values in Fig. 3 were inferred from the observed water levels through rating curves: depending on the rating curve applied, the maximum recorded water level (11.77 m) corresponded to an estimated discharge ranging between 750 m³/s and 840 m³/s. Conversely, the hydrological model, which did not account for the levee breaches that occurred between the Lamone and Marzeno confluence and the Reda station, estimated a peak discharge approaching 1000 m³/s.

5.  **Was the contribution of the downstream portion of the catchment (between Reda and the river mouth) included in the hydraulic model? If so, how was this inflow injected (e.g. at punctual cells as inlets located in another hydrological estimation point or spatially distributed throughout the river)? If not, would it be feasible to assess its impact on peak water levels near the levee breach, or conduct a sensitivity analysis to evaluate its influence on the simulated flood extent?**

    We thank the Referee for this comment. The downstream portion of the catchment was not included in the hydraulic model since neither minor channels nor tributaries flow into the Lamone River. To clarify this, we have modified the revised paper as follows (page 6 lines 132-134):

    *"Since neither minor channels nor tributaries flow into the Lamone River downstream of the town of Faenza, no additional inflows were considered in the hydraulic modelling."*

6.  **From Figure 3, it appears that sub-catchments were defined based on tributary confluences. A short clarification in the text would help confirm this assumption.**

We thank the Referee for this suggestion. The sub-catchments were delineated using a digital elevation model (DEM) and the hydrologic analysis tools available in ArcGisPro software. We have modified the revised paper at page 5, lines 107-110 as follows:

*"Meteorological forcing was assumed to be uniformly distributed within each sub-catchment, whose definition was delineated using a digital elevation model and the hydrologic analysis tools available in ArcGisPro software to determine flow direction, calculate flow accumulation, delineate watersheds, and create stream networks."*

7. **It would be useful to include a performance indicator (e.g. NSE, KGE, etc) for the hydrological model during both the calibration period and for the two extreme events. While the latter might be subject to higher uncertainty, such metrics would still provide useful insights into the model's overall behaviour.**

We thank the Referee for having raised this interesting point. The calibration of the hydrological model was quantitatively assessed by using the Nash-Sutcliffe Efficiency indicator. We have clarified this in the revised paper at page 6, lines 124-130:

*"The calibration of the hydrological model was quantitatively assessed by using the Nash-Sutcliffe Efficiency indicator defined as:*

$$NSE = 1 - \frac{\sum_{i=1}^{N}(Q_{m,i} - Q_{o,i})^2}{\sum_{i=1}^{N}(Q_{o,i} - \bar{Q}_o)^2} \qquad (1)$$

*where $Q_{m,i}$ is the ith value of the modelled discharges, $Q_{o,i}$ is the ith value of the observed discharges, and $\bar{Q}_o$ is the mean value of the observed discharges.*

*As shown in Fig. 3, the hydrological model well reproduced the 2008-2024 discharge series and even the September 2024 flood wave, which represented the second most severe event ever occurred in this station: the NSE indicator resulted equal to 0.83, further demonstrating the model's performance."*

**Technical Corrections:**

**Lines 22–23: Please consider adding the publication year to the ARPAE reference, e.g., (ARPAE, 2024).**

Thank you. This has been corrected.

**Lines 43–45: Consider expressing rainfall amounts only in [mm] and specify the associated catchment area. This would improve readability and reduce the need for conversions between [mm] and [m³].**

Thank you. This has been modified in the revised paper as follows:

*"Over these two days, the river basin was affected by a rainfall amount of about 270 mm, with 320 mm concentrated within an area of around 270 km$^2$ defined by the 280 mm isohyet (Fig. 1): the average precipitation depth in the central portion of the basin was approximately 1.8 times higher than the mean precipitation over the eastern part of the region."*

**Line 96: The phrase "spatially explicit" might be misleading. If the intention was to describe the numerical scheme, perhaps "explicit temporal scheme" was meant instead?**

Thank you. By "spatially explicit" we mean a model that simulates the hydrological cycle in detail, considering the spatial distribution of parameters such as rainfall, infiltration, evapotranspiration and runoff in the individual units (sub-basins) into which the basin has been divided. The expression does not refer to the numerical scheme adopted.

**Line 153: The phrase "fairly agree" should be revised for clarity. Consider alternatives such as giving an actual number or performance indicator.**

Thank you. This has been modified in the revised paper as follows:

> *"The simulated flooded area closely matches the actual one and the comparison between the water levels predicted by the model and those observed at three gauge stations fairly agrees in terms of timing and pick values (at Reda, Pieve Cesato and Mezzano stations, the misfits relative to the peak values are lower than the 5%, 2% and 9%, respectively)."*

**Line 153: Missing 's' in ' ... three gauge stations fairly agrees'.**

Thank you. This has been corrected.

**Line 167: Missing 's' in '... two intense and consecutive rainfalls'.**

Thank you. This sentence has been rewritten in the revised paper.

**Figure 3 : The sub-catchment subdivision should be realised in another colour and clearer. On a print version of the paper, the contours are barely visible.**

Thank you. The sub-catchment panel has been modified.